# The Influence of a Stressful Microenvironment on Tumor Exosomes: A Focus on the DNA Cargo

**DOI:** 10.3390/ijms21228728

**Published:** 2020-11-19

**Authors:** Rossana Domenis, Adriana Cifù, Francesco Curcio

**Affiliations:** 1Dipartimento di Area Medica, Università degli Studi di Udine, 33100 Udine, Italy; rossana.domenis@uniud.it (R.D.); adrieali@hotmail.it (A.C.); 2Istituto di Patologia Clinica, Azienda Sanitaria Universitaria Integrata Friuli Centrale, 33100 Udine, Italy

**Keywords:** tumor microenvironment, exosomes, extracellular vesicles

## Abstract

Exosomes secreted by tumor cells, through the transport of bioactive molecules, reprogram the surroundings, building a microenvironment to support the development of the tumor. The discovery that exosomes carry genomic DNA reflecting that of the tumor cell of origin has encouraged studies to use them as non-invasive biomarkers. The exosome-mediated transfer of oncogenes suggested a new mechanism of malignant transformation that could play a role in the formation of metastases. Several studies have examined the role of tumor exosomes on the modulation of the tumor microenvironment, but relatively few have been directed to assess how stressful stimuli can influence their production and cargo. Understanding the changes in exosome loads and the production pattern of the stressed tumor cell may uncover actionable mechanisms responsible for tumor progression.

## 1. Introduction

Exosomes are a population of nanometer-sized extracellular vesicles (EVs) that originate from the multivesicular endosome and are released from all cells. Their secretion was first described as a mechanism for the elimination of excessive proteins or undesirable molecules from the cell [1] and for many years, exosomes have been ignored by the scientific community. However, decades of research have identified exosomes as special couriers with a key role in the intercellular communication system involved in several physiological and pathological conditions [2]. Actually, exosomes contain specific active biological molecules such as proteins, lipids, and nucleic acids, suggesting the existence of a regulated mechanism that controls the selection of molecules to be transported by them [3].

Exosomes are released by tumor cells and their role in tumor progression and metastasis has been largely investigated. Tumor exosomes, reflecting the content of their cell of origin, can transfer oncogenic molecules to recipient cells that will be functionally influenced by them [4]. Evidence also suggests that tumor cells use exosomes to shape the microenvironment, making it favorable for their growth and suitable for the formation of pre-metastatic niches [5], promoting the immune-escape of tumor cells [6] and inducing extracellular matrix remodeling and angiogenesis [7]. It should be noted that tumor exosomes seem to “address” specific target sites in different organs by virtue of the expression of a specific integrin associated with the formation of pulmonary or hepatic metastases [8].

In recent years, several studies have examined the role of tumor exosomes on the modulation of the tumor microenvironment, but relatively few have been directed at assessing how stressful stimuli can affect their production. This review will discuss the influence of the tumor microenvironment on the composition of the exosome, focusing in particular on DNA content and its possible role in promoting malignant transformation.

## 2. Tumor Microenvironment Modulate the Release of Exosomes

It has been described that the molecular composition of exosomes, dictated by local microconditions and the stress state related to the tumor microenvironment, can induce an increase in the release of exosomes from cells and changes in the composition of their cargo [9,10] (Table 1). Cell culture conditions affect the amount of exosomes released from HeLa cells, which produce more exosomes when grown in 3D than in the 2D culture system [11]. Oxidative stress and the loss of Ca^2+^ homeostasis contribute to endoplasmic reticulum (ER) dysfunction due to the accumulation of incompletely or incorrectly folded proteins. Exosomes released from ER-stressed liver cancer cells mediate the JAK2/STAT3 activation pathway and promote the release of cytokines by macrophages [12].

Melanoma-derived EVs secreted under cytostatic, heat, and oxidative stress conditions show substantial differences in the miRNome and proteome, which results in several abnormalities in the recipient mesenchymal stem cells [13].

It is known that DNA damage, irradiation, and activation of oncogenes can cause cellular senescence by the acquisition of a pro-inflammatory phenotype known as the senescence-associated secretory phenotype (SASP). EVs secreted by senescent cells appear to be a new SASP factor that can exert pro-tumorigenic activity for pre-malignant recipient cells [14,15,16]. Recent studies have shown that ionizing radiation induces increased release of exosomes [17,18] and changes in their composition with the overexpression of proteins involved in transcription, translation, protein turnover, cell division, and signaling [19].

It has been noted that the chemotherapeutic drug Paclitaxel may induce senescence in triple negative cancer cells accompanied by increased secretion of exosomes [20]. Senescence not only increases EVs secretion, but also alters their cargo. EVs secreted by fibroblasts and senescent epithelial cells bind to ephrin-A1, a molecule highly expressed in several types of cancer cells, and which promotes the proliferation of cancer cells through the reverse signaling of EphA2/ephrine-A1 [21]. In addition, senescent stromal cells release more exosomes which, in turn, are able to increase chemotherapy resistance in receiving cancer cells in a mechanism that appears to involve the silencing of SIRT1 [22].

Hypoxia is an important feature of the tumor microenvironment. Tumor growth in low oxygen conditions exhibits more aggressive phenotypes and is associated with a poor patient outcome in a wide variety of cancers [23]. Several studies have clearly indicated that hypoxic stress promotes release and induces qualitative changes in the exosomes secreted by cancer cells [24,25], which influence the metastatic potential of normoxic cells. Mass spectrometry analysis of exosomes derived from glioblastoma cells kept under hypoxic conditions detects many of the major players regulating angiogenesis, such as TSP-1, LOX, VEGF, and ADAMTS1 [26]. Exosomes from hypoxic nasopharyngeal carcinoma are enriched in MMP13 and promote cell migration and invasion of normoxic cells [27]. Similarly, vesicles secreted from hypoxic colorectal cancer cells are enriched in miR-410-3p, which transferred into the normoxic cells increased their metastatic potential through activation of the PI3K/Akt signaling pathway [28]. Exosomes derived from glioblastoma (GBM) cells grown in hypoxic compared to normoxic conditions are powerful inducers of ex vivo and in vitro angiogenesis through phenotypic modulation of endothelial cells. HUVECs and HBMECs cells were induced by hypoxic exosomes to secrete cytokines and growth factors and stimulated primary pericyte PI3K/AKT signaling activation and migration. Moreover, exosomes derived from hypoxic compared with normoxic conditions showed increased autocrine pro-migratory activation of GBM cells and accelerate tumor expansion in a mouse GBM xenograft model [29]. Exposure of human breast cancer cells to hypoxia stimulates the expression of the RAB22A HIF-dependent gene and increases the shedding of exosomes. In addition, incubation of naïve breast cancer cells with exosomes shed by hypoxic breast cancer cells promotes focal adhesion formation, invasion, and metastasis [30].

Hypoxia results in acidification of the tumor microenvironment, which may have a profound influence on the release and uptake of exosomes. The microenvironmental acidification of the melanoma cell culture conditions increases the intra-tumor production and uptake of exosomes, which leads to activation of the metabolic pathways related to tumor aggressiveness [31].

Chronic inflammation plays a key role in the progression of cancer and in the development of interactions between host and tumor cells. Increasing evidence indicates that inflammasome activity is correlated with higher EVs production and modulation of their protein cargo [32]. Stimulation of tumor cells with inflammatory cytokines induces qualitative and quantitative changes in the exosomal proteome, many of which are involved in cancer progression. Pro-inflammatory IL-1β and TNF-α cytokines result in changes in the exosomal proteome of U373 glioma cells, significantly increasing the levels of CRYAB (HspB5), a heat shock protein with anti-apoptotic activity [33]. Activation of toll-like receptor 4 (TLR4) induces an inflammatory signal that increases the tumorigenic potential of cancer cells and promotes their immune evasion, stimulating the release of more effective immunosuppressive exosomes. Exosomes released from tumor cell lines following activation of TLR4 by LPS modulate the production of inflammatory proteins by monocytes, influence the expansion regulatory T cells, and reduce the expression of NKG2D on CD8+ T cells [34].

**Table 1 ijms-21-08728-t001:** Effects of stressful microenvironment on extracellular vesicles (EVs).

**EVs Source**	**Stress**	**Effect on EVs**	**REFs**
Melanoma	Cytostatic, heat and oxidative	miRNome and proteome changes	[13]
Keratinocytes	Ionizing radiation	Increasing release	[17]
Prostate cancer	Radiation	Enrichment in B7H3 protein content	[18]
Head/neck squamous carcinoma	Ionizing radiation	Enrichment in cell signaling proteins	[19]
Breast cancer	Chemotherapeutic drug	Increasing release	[20]
Retinal pigment epithelial	DNA-damaging agent	Increasing release, pro-tumorigenic	[21]
Breast cancer	Hypoxia	Increasing release	[24]
**EVs Source**	**Stress**	**Effect Induced by EVs**	**REFs**
Liver cancer	Oxidative	Pro-inflammatory	[12]
Glioblastoma	Hypoxia	Pro-tumorigenic	[26]
Nasopharyngeal carcinoma	Hypoxia	Pro-tumorigenic	[27]
Colorectal cancer	Hypoxia	Pro-tumorigenic	[28]
Glioblastoma	Hypoxia	Pro-tumorigenic	[29]
Breast cancer	Hypoxia	Pro-tumorigenic	[30]
Melanoma	Acidification	Pro-tumorigenic	[31]
Glioblastoma	Inflammation	Anti-apoptotic	[33]
Cancer cell lines	Inflammation	Immunosuppressive	[34]

## 3. Tumor Exosomes Transport DNA

The production of exosomes was initially proposed as a mechanism to maintain cellular homeostasis by removing excess or obsolete molecules from cells, but further studies have attributed them to a sophisticated role in cell–cell communication. Recently, it has been shown that exosomes, in addition to several types of nucleic acids (mRNA, miRNA, IncRNA, and viral RNA), transport mitochondrial [35,36] and single and double-stranded genomic DNA (dsDNA) [37]. The analysis of nucleic acid content of small EVs separated using high-resolution iodixanol density gradients demonstrated that most of the DNA was found in the high-density fractions. The authors defined these vesicles as “non-canonical exosomes”, as they contained histone proteins and other non-vesicular proteins [38].

Exosomes allow the elimination of damaged DNA accumulated in the cytoplasm and as proof of the importance of this function, it has been shown that inhibition of their production promotes cellular senescence by stimulating the cGAS/STING inflammatory pathway [39]. Additionally, exosomes derived from several types of cancer transport a greater amount of DNA [40], although to date, it has not yet been clarified whether the mechanism of DNA sorting in tumor exosomes is random or specifically regulated.

Using whole genome sequencing, it has been shown that serum exosomes from patients with pancreatic cancer contain > 10 Kb fragments of genomic dsDNA spanning all chromosomes [41]. DNase treatment of supernatant exosomes of cultured cells has revealed that dsDNA is present both inside the exosomes and associated with their surface [40,42,43]. In addition, retrotransposon elements were found in the DNA transported by the exosomes derived from tumor cells [44]. 

## 4. Tumor Exosomal DNA as Diagnostic Biomarker

The discovery of exosomal DNA has attracted considerable attention as a non-invasive source for early diagnosis of cancer and monitoring of treatment response. Exosomal DNA could be used as a genetic marker for the detection of mutations, since the same mutations as the parent cancer cells are found in the transported DNA. Exosomes are present in blood and other biological fluids in large numbers and cancer cells generally release many exosomes continuously. It has been estimated that more than 90% of cell-free DNA (cfDNA) in human blood is found in plasma exosomes [45]. On the other hand, a recent study has shown that large EVs (oncosomes) isolated from prostate cancer patient plasma by differential centrifugation contained more DNA with respect to small EVs (exosomes), despite exosomes being more numerous [46].

Exosomal DNA is stable under several storage conditions because its encapsulation provides it with enhanced stability than when it is outside the cells [47]. On the other hand, the diagnostic value of exosomal DNA remains to be clarified due to the challenges associated with detection sensitivity/specificity [48] and signal-to-noise ratio due to the mixing of DNA fragments from tumor and normal cells.

Several studies have documented the presence of dsDNA in the exosomes analyzed in supernatants of various cancer cell lines (melanoma, breast, lung, prostate, pancreatic cancers) [40]. 

The detection of EGFR mutations in exosomes isolated from malignant pleural effusions [49,50] and KRAS and BRAF mutations in exosomes isolated from serum of patients with colorectal cancer [51] shows elevated sensitivity, specificity, and coincidence rate compared to tumor tissue analysis. The exosomal DNA isolated from ascites from patients with ovarian cancer reflects the copy number variation (CNV) status of the primary tumor [42]

It is important to note that the analysis of exosomal DNA has become particularly relevant when it is difficult to detect mutations prior to surgery, such as for neuronal tumors [52,53]. Serum exosomal dsDNA from patients with pheochromocytomas (PCCs) and paragangliomas (PGLs) shared the same mutations in ET, VHL, HIF2A, and SDHB genes with that of the parent tumor cells. The excellent concordance rates (97.6–100%) between exosomes and tumor tissue support the use of exosomal DNA as a noninvasive genetic marker compared to liquid biopsies obtained from cerebrospinal fluid (CSF) [52].

All types of EVs (apoptotic bodies, shedding microvesicles, and exosomes) secreted by glioma cells in a xenograft mouse model cross the intact blood–brain barrier into the bloodstream. In addition, glioma-derived EVs could be detected in the peripheral blood of a small cohort of low and high grade glioma patients, regardless of the BBB integrity. The presence of IDH1^G395A^ sequences, an essential biomarker in the current management of human glioma, was detected by conventional PCR in 80% of low-grade gliomas samples, matching results from IHC and conventional PCR on solid samples [54].

## 5. Mutated DNA of Tumor Exosomes Promotes Malignant Transformation

The exosomal DNA, as for other molecules contained in them, can be transferred from one cell to another (same cell type or even different) and translocated in the nucleus. EVs secreted by marrow-derived mesenchymal stromal cells (BM-hMSC) transfected with an exogenous *Arabidopsis thaliana*-DNA (*A*.*t*.-DNA) mediate the horizontal transfer of DNA into recipient cells. Sequencing analysis revealed that exogenous DNA was stably integrated into the genome of recipient cells, even if authors specify that this was a rare event [43].

It has been proposed that horizontal DNA transfer from tumor cells to normal cells represents an important mechanism for tumorigenesis, tumor progression, and metastasis (Table 2). Oncogenic DNA sequences transported by EVs accumulate in circulating neutrophils, which represent the major reservoir of this material in the blood of mice harboring H-*ras*-driven xenografts. Tumor excision resulted in the disappearance of the leukocytes-associated gDNA signal within several days, which is in line with the expected half-life of these cells. In addition, the exposure of HL-60 myeloid cells to EVs from H-*ras*-driven cancer cells resulted in an increase in tissue factor procoagulant activity and interleukin 8 production [55].

Circulating DNAs isolated from the blood of cancer patients are deleterious for normal cells as they can damage their DNA, integrating into the genome, and activate proteins of the DNA damage–repair response and apoptotic pathways. Fragmented DNA was readily taken up by culture cells, localized in their nuclei within minutes and the genomic integration, detected by FISH and NGS on recipient cells, led to dsDNA breaks, measured as activation of H2AX [56].

The presence of active retrotransposon elements in the genetic material of exosomes suggests a possible role in the formation of genetic instability in recipient cells [57]. Microarray analysis of cellular RNA and exoRNA sequences from glioblastoma primary cells indicated higher transcription levels of several retrotransposon sequences, including human endogenous retroviruses (HERV), Line-1 (L1), and ALU elements in microvesicles with respect to donor cells [44]. EVs released by H-*ras* transformed epithelial cells that contain dsDNA fragment covering the entire genome, including full length H-*ras* oncogene sequences which can be transferred to non-tumorigenic fibroblasts. H-*ras* signals were readily detected by PCR within 2 days of EV exposure, and were still present as long as 30 days later, albeit at diminished levels, with concomitant mitogenic response of recipient cells [58].

To date, it is not yet clear whether EVs containing oncogenes exert regulatory rather than transforming influence on recipient cells. Primary and immortalized fibroblasts are susceptible to phenotypic transformation induced by H-*ras*-EVs, but these effects are transient and do not produce a permanent tumor transformation of these cells, even after several months of observation [59].

On the other hand, cancer cell exosomes have been shown to contribute to malignant transformation when used as an initiator in combination with a promoter (TPA), which induce cell proliferation in the classic two-step cell transformation assay. Exosomes released from pancreatic cancer-derived cell lines, unlike those produced by normal pancreatic cells, introduce mutations in NIH/3T3 recipient cells that under proliferative stimulation, completely transform and induce the formation, in culture, of foci capable of inducing tumors when injected into mice [60]. It was possible to transfer the hybrid BCR/ABL gene, involved in the pathogenesis of chronic myeloid leukemia (CML), to neutrophils from EVs derived from K562 in an in vitro model [61]. Injection via tail of K562-derived EVs caused BCR/ABL mRNA and protein de novo synthesis and induced CML in NOD/SCID mice, demonstrating that DNA transfer through EVs can have pathogenetic significance [62].

**Table 2 ijms-21-08728-t002:** Effect of Exo-DNA transfer.

Maternal Cells	Recipient Cells	Effect of Exo-DNA Transfer	REFs
Cancer-associated fibroblasts	HTS/HTD breast cancer	Resistance to therapy	[36]
H-*ras*-driven cancer cells	Neutrophils	Stimulation of procoagulant and proinflammatory activity	[55]
H-*ras* transformed RAS-3 cells	Epithelial RAT-1 cells	Increasing of cells proliferation	[58]
H-*ras* transformed RAS-3 cells	Epithelial RAT-1 cells	Malignant transient transformation	[59]
Pancreatic cancer cells	NIH/3T3	Malignant transformation	[60]
K562	Neutrophils	Decreasing of phagocytic activity in vitro	[61]
K562	Neutrophils	Decreasing of phagocytic activity in vivo	[62]

## 6. Emerging Role of Tumor Microenvironment in Modulating the Sorting Mechanism and Transformation Potential of Exosomal DNA

The mechanism for loading DNA into exosomes remains poorly understood. The formation of micronuclei (MN) in cancer cells, following the induction of genomic instability with genotoxic drugs, promotes the release of DNA-carrying exosomes by ovarian cancer cells. Collapsing of MN directly interacts with the molecular machinery for exosome biosynthesis via tetraspanins, suggesting that the cargo of disrupted MN is loaded into exosomes [42]. On the other hand, the presence of mutant DNA sequences in tumor exosomes does not seem to be sufficient to induce malignant transformation in recipient cells and the role of the microenvironment in promoting cells proliferation could be determinant. Exosomes derived from pancreatic cells are able to induce genetic changes in recipient cells even if they are unable to completely transform them on their own [60]. The effect of the tumor microenvironment on the production and composition of exosomes could explain the apparent contradiction between studies, showing that transformation induced by oncogenic exosomes is transient [58,59] and those showing a pathogenetic significance of tumor genes are transferred in vivo [60,61,62].

Our preliminary data suggested that the activation of TLR4 modulates the packaging of mutated genes in exosomes, which are efficiently transferred into normal recipient cells. We observed that exosomes derived from SW480 cells are able to transfer TP53 dsDNA fragments and KRAS-mutated sequences, which are efficiently internalized by CCD841 colon epithelial and THL-2 hepatocytic cells. Copies of mutated DNA and relative mRNA were found in recipient cells up to one month after discontinuation of treatment with exosomes. Note that the activation of TLR4 with LPS stimulates the packaging of the mutated DNA in the exosomes and consequently, more mutated DNA was found in the recipient cells treated with exosomes released from cells activated by TLR4 stimulation [63].

## 7. Concluding Remarks

The study of the tumor microenvironment is improving our knowledge of the underlying mechanisms of tumor formation, progression, and metastasis. Exosomes play a key role in the modulation of the microenvironment by orchestrating changes on behalf of tumor cells. Oncogenic gDNA sequences are transported by the tumor exosomes and are delivered through the bloodstream to sites distant from the primary tumor location. There, exosomes could be internalized by receiving cells and induce a malignant transformation. In this process, microenvironmental stressors could play a crucial role in promoting DNA sorting within the exosomes (Figure 1). Further studies are needed to investigate this possible and innovative mechanism of exoDNA-mediated cancer pathogenesis.

## Figures and Tables

**Figure 1 ijms-21-08728-f001:**
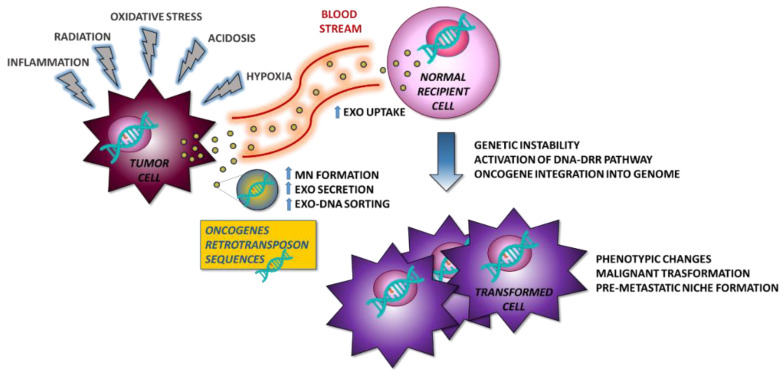
Proposed model for the formation of metastases. Tumor cells under stress conditions secrete exosomes, which transport oncogenes capable of inducing malignant transformation in normal recipient cells and contribute to the formation of the pre-metastatic niche also controlling the immune environment. MN—micronuclei.

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
