# Peer review of "The Influence of a Stressful Microenvironment on Tumor Exosomes: A Focus on the DNA Cargo"

_ijms, 2020, doi:10.3390/ijms21228728_

Round 1

Reviewer 1 Report

Dear Editors,

The authors have satisfactorily addressed all the comments raised during the first review. I recommend that this revised manuscript should be accepted for publication.

Thank you,

Sincerely,

Ravi P. Sahu, Ph.D.

Author Response

We thank the reviewer for his positive opinion.

Reviewer 2 Report

The review deals with an important issue regarding the effect of stressful stimuli on tumor exosomes, with a specific focus on the DNA cargo.

The review is well written and the topic interesting. I have only few concerns:

  • The authors should better explain why they decided to focus only on exosomes, excluding from their analysis large extracellular vesicles. Several reports (e.g. doi.org/10.1080/20013078.2018.1505403) indicate the important contribution of tumor-derived large EVs carrying most of the DNA. The authors could evaluate the possibility to extend their analysis also on large EVs.
  • In the context of tumor vesicles and DNA, the authors should cyte some pivotal papers, such as doi.org/10.1080/20013078.2019.1656993, indicating how different subfractions can be distinguished by their DNA content. This could further  improve the manuscript.

Author Response

We thank the reviewer for the suggested references that have been discussed in the manuscript.